# Non-Separability of Physical Systems as a Foundation of Consciousness

**DOI:** 10.3390/e24111539

**Published:** 2022-10-26

**Authors:** Anton Arkhipov

**Affiliations:** MindScope Program, Allen Institute, Seattle, WA 98109, USA; antona@alleninstitute.org

**Keywords:** consciousness, integrated information theory, non-separability, classical limit, quantum systems

## Abstract

A hypothesis is presented that non-separability of degrees of freedom is the fundamental property underlying consciousness in physical systems. The amount of consciousness in a system is determined by the extent of non-separability and the number of degrees of freedom involved. Non-interacting and feedforward systems have zero consciousness, whereas most systems of interacting particles appear to have low non-separability and consciousness. By contrast, brain circuits exhibit high complexity and weak but tightly coordinated interactions, which appear to support high non-separability and therefore high amount of consciousness. The hypothesis applies to both classical and quantum cases, and we highlight the formalism employing the Wigner function (which in the classical limit becomes the Liouville density function) as a potentially fruitful framework for characterizing non-separability and, thus, the amount of consciousness in a system. The hypothesis appears to be consistent with both the Integrated Information Theory and the Orchestrated Objective Reduction Theory and may help reconcile the two. It offers a natural explanation for the physical properties underlying the amount of consciousness and points to methods of estimating the amount of non-separability as promising ways of characterizing the amount of consciousness.

## 1. Introduction

What are the physical foundations of consciousness? This question, perhaps posed in a less modern way, occupied philosophers for millennia and motivated an increasing amount of scientific research in recent years. While the definition of what should be called consciousness is still sometimes debated, here we will focus on the basic definition of consciousness as the phenomenon of having a subjective experience [1,2]. It is what we have during normal waking state or when dreaming during sleep, such as seeing a visual scene, experiencing an emotion, thinking of something. It is also the definition that is at the heart of the “hard problem” of consciousness [3]—why do we experience anything at all, and what determines the ability of some physical systems such as brains—and, presumably, not others, such as rocks—to have subjective experiences?

To answer this question, we ultimately need a theory of how consciousness arises from physical constituents of the Universe, such as fields, particles, and their interactions. Two major theories in this space attracted much attention in recent decades: the Integrated Information Theory (IIT) of Giulio Tononi and colleagues [4,5,6] and the Orchestrated Objective Reduction (Orch OR) theory of Penrose and Hameroff [7,8,9,10,11]. The IIT starts from postulating several essential properties of phenomenal experience and derives from them the requirements that must be met for physical systems to have consciousness. In the resulting framework, consciousness reflects the intrinsic cause-effect power of the system’s components upon themselves as a whole (i.e., beyond the sum of the parts), and is quantified by the “integrated information” measure, also referred to as “Phi”. Orch OR, on the other hand, proposes that consciousness arises from ‘orchestrated’ coherent quantum processes (proposed to occur within microtubules in the brain’s neurons), where the continuous quantum evolution of each process terminates via the “objective reduction” of the quantum state. Each moment of such reduction of the uncertainty of the quantum state to a 100% certain classical realization corresponds in Orch OR to moments of consciousness.

Many other proposals exist, especially with respect to the brain mechanisms and computational principles involved in consciousness, such as the Global Workspace Theory [12,13] and Global Neuronal Workspace Theory [14,15,16], the Temporo-spatial Theory of Consciousness [17], the Higher-Order Theory of consciousness [18,19,20,21], Attention Schema Theory [22,23], the Free Energy principle [24,25], the Information Closure Theory [26], and others. See Refs. [27,28] for a comprehensive account. These theories are interesting and important but focus more on the neural correlates of consciousness [2,29,30,31] or cognitive aspects and psychology of consciousness, rather than the basic physical underpinnings of consciousness as a physical phenomenon, or, as argued by Tegmark [32], as a state of matter. See also the discussions by Northoff and colleagues of the neural predisposition of consciousness (NPC) [17,33,34,35], which are the biophysical and physiological necessary conditions of possible consciousness.

Here we consider the phenomenon of consciousness from such a first-principles, physics-based point of view. What are the fundamental physical characteristics that determine whether consciousness exists, and in which amount, in any physical system? Existing proposals offer possible answers (though we do not know at present whether true or not), but also leave some questions open.

For example, the IIT [5] takes a point of view that one may call ‘computational’, as it typically operates with systems composed of logical gates with discrete states (e.g., “on” and “off”) and connections that determine logical interactions. This is a valid approach, but can a theory of consciousness be derived from ‘regular’ physics operating with Hamiltonians, particles, and fields, with both continuous (e.g., position) and discrete (e.g., spin) degrees of freedom? After all, the brain, the only system for which we know with some certainty that it generates consciousness, is a piece of physical matter consisting of interacting particles such as protons, neutrons, and electrons. If we have a Hamiltonian and a wave function of all the degrees of freedom in the brain, which is all there is according to the physics view, where does consciousness enter? Perhaps a theory answering this question can even turn out to be equivalent to IIT, once correspondence is established between the Hamiltonian-plus-wave-function view and the logical-gates-plus-connections view (similar to matrix-based and wave function-based formulations of quantum mechanics being equivalent to each other).

The Orch OR theory [11] answers the question above by suggesting the objective reduction of the wave function as a fundamental consciousness mechanism, though this appears to require quantum coherence over the length scale of the whole brain (~1–10 cm) and time scale of a ‘conscious percept’ (typically assumed to be ~50–100 ms [36,37,38]). Though not impossible in principle, it remains to be seen whether such quantum phenomena operate on this scale in the brain; it is perhaps fair to say that the majority opinion at present is that the brain functions mostly in the classical limit. Furthermore, these difficulties aside, the Orch OR theory typically focuses on how consciousness comes to be, but one also needs to know how to quantify the amount of consciousness and its composition in the system. For example, when the wave function collapses according to the Orch OR recipe, which of the brain’s degrees of freedom are part of this process and which are not, and how does that matter for the conscious experience?

Here we hypothesize that *non-separability* of the state of a physical system is the fundamental property determining the presence and amount of consciousness. In simple terms, non-separability of some degrees of freedom in a physical system means that these degrees of freedom cannot be described, with regard to their state and time evolution, as independent variables, but form a truly inseparable complex that is not simply a sum of its parts. Mathematically, non-separability means that the function describing the time evolution of a system cannot be represented as a product of functions describing the system’s parts. For example, if the wave function describing a system of two degrees of freedom, $\psi \left(x_{1},x_{2},t\right)$, can be written as the product of the wave functions describing each degree of freedom, $\psi \left(x_{1},x_{2},t\right)=\psi_{1}\left(x_{1},t\right)\psi_{2}\left(x_{2},t\right)$, then this system is separable, and if it cannot ($\psi \left(x_{1},x_{2},t\right)\neq \psi_{1}\left(x_{1},t\right)\psi_{2}\left(x_{2},t\right)$), then it is non-separable. This offers a natural and general physics-based mechanism for the *integration* of a subset of components from a system into a ‘whole’ subsystem, that is internally united and separate from the rest of the system, which is a hallmark of consciousness.

We propose that such non-separable complexes are conscious, and the extent to which they are conscious is determined by the extent of non-separability and the number of degrees of freedom involved. This determines the amount of consciousness each complex has. By amount of consciousness, we mean the measure of consciousness, a hypothetical number that characterizes consciousness and possibly its richness, such as Phi in IIT. We discuss this in more detail in Section 2.9 and point out that multiple different measures may be useful or even necessary to characterize different aspects of consciousness, although we expect the degree of non-separability in a system to be a key measure quantifying its consciousness. While mathematically the non-separability concept appears simple, in practice it is anything but, and below we describe a number of observations following from our hypothesis and make suggestions for further inquiry in this area. The hypothesis appears consistent with both the IIT and Orch OR, suggesting that the fundamental property of non-separability may be important to consider in attempts to refine, unify, and test theories of consciousness.

## 2. Results

### 2.1. Hypothesis: The Amount of Consciousness in a Physical System Corresponds to the Extent to Which This System Is Non-Separable

Here we use the definition of consciousness as the phenomenon of having a subjective experience [1,2], such as seeing something, having a thought, or experiencing an emotion. Clearly, human conscious experiences are often highly complex, combining perception of various sensory modalities and inner states at the same time and occurring at multiple spatio-temporal levels. Our task here is not to uncover the nature of all the possible aspects of such experiences, but rather to understand how the basic phenomenon of consciousness—the first-hand experience, or feeling anything at all—comes about.

If we want to explore physical properties that form the basis of consciousness, it is useful to keep in mind as guiding goalposts the basic attributes of consciousness. Such attributes have been enumerated as five features of consciousness forming the foundation of IIT [5] (originally called “axioms”, although objections have been raised to calling them such [39]), and we will repeat them here. (1) The intrinsic existence feature: A conscious system must be able to change its state due to interactions of its components. (2) The composition feature: Experience is composed of multiple components within it. (3) The information feature: Any experience is specific, i.e., distinct from other experiences due to a particular set of components combined in that experience but not others. (4) The feature of integration: All components of a conscious experience are bound in a single whole—the irreducibility of a conscious state that is quantified by the “integrated information” measure, Phi. (5) The exclusion feature: Experience is definite in content, such that certain components of the system are ‘in’ the conscious complex, and others are not, and likewise this complex exists on a certain time scale, and not faster or slower.

One may suggest that features (1–3) are satisfied by a large class of physical systems, where system components have sufficient interactions with each other to change the overall state; where the system’s state is determined by the states of multiple heterogeneous components; and where at least some distinct states of the system are non-degenerate in the sense that they correspond to distinct configurations of the system’s components and affect the system’s dynamics differently. However, features (4) and (5) require some additional principle that would ensure integration of the system’s components into a ‘whole’ and establishing a boundary between what is ‘in’ and what is ‘out’ of the conscious complex.

We propose that non-separability is such a principle. Non-separability means that certain degrees of freedom in a system cannot be described as independent variables but need to be considered together as a ‘whole’. This directly establishes the integration and a boundary, per features (4) and (5) above. Some degrees of freedom may be separable from the rest, so they would be ‘out’. The rest are ‘in’, integrated together by virtue of a particular structure of their interactions with each other. Importantly, non-separability can apply to both continuous and discrete degrees of freedom.

Note that determining whether certain degrees of freedom in a system are separable or not in a general case is a hard, unsolved problem, and a simple presence of interactions or correlations between degrees of freedom is not sufficient for non-separability, as we illustrate below. Characterizing separability or non-separability in a general case, also known as the state factorization problem, has been highlighted by Tegmark as highly relevant for the problem of consciousness [32], though, to our knowledge, a specific relationship between non-separability and consciousness such as the one proposed here has not yet been established in the literature.

The hypothesis we posit here is that non-separability is equivalent to consciousness, i.e., it is necessary and sufficient for consciousness, and a conscious experience is what it feels to be a non-separable system. In other words, when multiple degrees of freedom are intertwined in a non-separable state, they form a multi-dimensional complex that conforms to the IIT features above and possesses experiences determined by its states in this multi-dimensional space. The more dimensions are intertwined and the stronger the non-separability of these dimensions, the higher is the amount of consciousness. As a consequence, one obtains that any non-separable system (of two degrees of freedom or more) has some amount of consciousness, which, however, is expected to be minuscule in most cases. Presumably, brains support extraordinarily large and well-organized non-separable states, which result in the type of human consciousness we are used to.

Below, we discuss this hypothesis in more detail, consider a number of informative examples, and speculate about possible measures for characterizing the amount of consciousness, as well as about the relationship of our hypothesis with the IIT and Orch OR. For our current purposes, we assume that physical systems are governed by the regular quantum mechanics (i.e., leaving aside field theory, relativistic effects, or gravity), including its classical limit.

### 2.2. The Non-Separability Concept and Consciousness

We hypothesize that non-separability is equivalent to consciousness in that the extent of non-separability and the number of degrees of freedom involved determine the amount of consciousness.

Consider a function, $f\left(x_{1},\;x_{2},\ldots ,t\right)$, that fully describes the time evolution of a system with degrees of freedom $x_{1},x_{2},\ldots$; t stands for time. Separability means that the function for the full system can be represented as a product of functions that depend on the different degrees of freedom, for example:$$f\left(x_{1},x_{2},\ldots ,t\right)=f_{1}\left(x_{1},t\right)f_{rest}\left(x_{2},\ldots ,t\right).$$

In this example, the degree of freedom $x_{1}$ is separable from the rest of the system, $x_{2},\ldots$. Another example is shown in Figure 1, where we consider 10 degrees of freedom. The whole system is separable, but subsystems consisting of the degrees of freedom $x_{3},x_{4},x_{5},x_{6},\;x_{7}$ and $x_{9},x_{10}$ are non-separable:$$f\left(x_{1},x_{2},x_{3},x_{4},x_{5},x_{6},\;x_{7},\;x_{8},x_{9},x_{10},t\right)=f_{1}\left(x_{1},t\right)f_{2}\left(x_{2},t\right)g\left(x_{3},x_{4},x_{5},x_{6},\;x_{7},t\right)f_{8}\left(x_{8},t\right)h\left(x_{9},x_{10},t\right).$$

According to our hypothesis, these two non-separable systems possess certain (very small) amounts of consciousness, presumably more for the system with five non-separable degrees of freedom than for the system with two degrees of freedom. (However, as we will discuss below, non-separability may be weaker or stronger depending on the interactions between the degrees of freedom, and therefore the exact relationship between the amount of consciousness and the number of non-separable degrees of freedom in a subsystem can be complicated.)

Thus, for any system, even the whole Universe, the function describing its state can potentially be decomposed into a product of functions that each describe subsets among all the degrees of freedom. The system is then decomposed into non-separable subsystems. Our hypothesis is that the non-separable subsystems form the units which have some amount of consciousness.

What is the appropriate function $f\left(x_{1},\;x_{2},\ldots ,t\right)$ that should be considered when separability and non-separability of a physical system is in question? Generally speaking, this is the function describing the time evolution of the state of the system, that is, the quantum-mechanical wave function $\psi \left(x_{1},x_{2},\ldots ,t\right)$ or density matrix, or the equivalents such as the Wigner function [40,41], the Marginal Distribution or Quantum Tomogram [42,43,44,45,46], etc. In a classical world, this can be the function that corresponds to the classical approximation of the wave function (or its equivalents). While any such function describes physical systems equivalently well, we propose below that the Wigner function may offer a particularly fruitful approach, as it has a direct equivalent in the classical case—the Liouville density function in phase space. Note that both in the quantum and classical case this must be a time evolution function of the system (such as, again, wave function, Wigner function, Liouville density function, etc.), rather than just the description of the instantaneous state. This means that it is not only the instantaneous state itself that matters, but the time evolution as well, reflecting the underlying Hamiltonian [32].

### 2.3. Non-Separability and the Exclusion Principle

The fifth feature (“axiom”) of consciousness posited by the IIT [5] is that of exclusion: certain components of the system are ‘in’ the conscious complex, and others are not. While the discussion above and the example in Figure 1 clearly establish that non-separability implies integration of the system’s components into a ‘whole’, one may ask whether they address exclusion itself. After all, in the IIT framework one needs to look for a combination of the system’s components that maximizes the integrated information Phi. In the example in Figure 1, the IIT requires that one computes Phi, e.g., for the two subsystems—one consisting of components {3, 4, 5} and the other of components {6, 7}—and compares that to the Phi of the subsystem of all five components {3, 4, 5, 6, 7}. If the latter gives the maximum of Phi, then {3, 4, 5, 6, 7} is the conscious complex and otherwise {3, 4, 5} and {6, 7} are two separate smaller conscious complexes. How does this concept figure in the non-separability hypothesis?

The answer is that the exclusion principle is embedded in the concept of non-separability. The non-intuitive aspect of it is that non-separability is what one establishes *after* all the possible partitions have been considered. In the example in Figure 1, one can, of course, ask whether the subsystem {3, 4, 5} is separable from {6, 7}. But, if it was, the subsystem {3, 4, 5, 6, 7} would not have been non-separable. In other words, if any component or combination of components of {3, 4, 5, 6, 7} is separable from the rest, then, clearly, {3, 4, 5, 6, 7} cannot be non-separable. Practically, establishing non-separability would indeed require considering all possible partitions, similar to how one would approach maximization of Phi in the IIT. This can be a very hard, computationally expensive procedure (see more on this below). However, the end result is that once non-separability of a system is established, by definition no subsystem within it can be described separately. If it could be, then the composite system would be separable, similar to how {1, 2, 3, 4, 5, 6, 7, 8, 9, 10} in Figure 1 is separable into {1}, {2}, {8}, {9, 10}, and {3, 4, 5, 6, 7}, but {9, 10} and {3, 4, 5, 6, 7} are not separable any further.

Thus, non-separability establishes both integration and exclusion.

### 2.4. Non-Interacting Systems Have Zero Consciousness

If non-separability is equivalent to consciousness, which physical systems have consciousness, and how much? To start addressing this question, let us first ask which systems have exactly zero consciousness by virtue of being strictly separable.

The simplest case is that of non-interacting systems, i.e., those where no degrees of freedom (which we also refer to as “components”, for convenience) interact with each other (Figure 2A). However, note that any component may be subject to an external potential. In this case, every component is completely independent of any other component, and such a system with no interactions between its parts is exactly separable [47], or factorizable:$$f\left(x_{1},x_{2},\ldots ,t\right)=f_{1}\left(x_{1},t\right)f_{2}\left(x_{2},t\right)\ldots =\prod_{k}f_{k}\left(x_{k},t\right).$$

According to our hypothesis, this system will then have no consciousness on its own, and each of its parts will likewise have zero consciousness.

Note that in principle it is possible to create a non-separable state (e.g., quantum-entangled—see more on the entanglement below) of a system and then effectively “turn off” interactions, after which the state may still remain non-separable. However, strictly speaking, this example does not correspond to the case above, where the Hamiltonian of the system never contains any interactions between the components.

Thus, we have shown that purely non-interacting systems are unconscious. It is important to note that in many cases interactions between components of a system may be weak but non-zero. Consider such systems that the strength of interactions between the components is controlled by a multiplicative parameter Q, so that for Q=0 all interactions are non-existent. Systems with high values of Q may have some amount of consciousness, but in the limit of Q approaching zero, interactions among the components disappear, and the system becomes separable and thus unconscious. This thought experiment suggests that, in the framework of our hypothesis, the amount of consciousness is best viewed as a continuous nonnegative number, rather than a binary one (i.e., “either consciousness is there or there is none”). In the former case, the amount of consciousness can take on infinitesimally small nonnegative values and approach zero as interactions become vanishingly small.

### 2.5. Interacting Systems Can Also Have Zero Consciousness

If no interaction means no consciousness, does it mean that any interacting system will have some amount of consciousness? It turns out the answer is no, as the following simple example shows.

Consider two particles interacting via the potential $v\left(r_{1}-r_{2}\right)$, where $r_{1},\;r_{2}$ are the coordinates of the particles (Figure 2B). Because of the interaction, the function describing this system in these original coordinates is non-separable:$$f\left(r_{1},r_{2},t\right)\neq f_{1}\left(r_{1},t\right)f_{2}\left(r_{2},t\right)$$

However, we can use a different set of coordinates,
$$R=\frac{r_{1}+r_{2}}{2},\;r=r_{1}-r_{2},$$
such that the Hamiltonian governing the evolution of this system now contains the potential energy term influencing only r, but not R.

Therefore, just as before with the non-interacting system, the function describing the system (as mentioned above, this is the time evolution function of the system) becomes separable:$$f\left(r_{1},r_{2},t\right)=f_{COM}\left(R,t\right)f_{rel}\left(r,t\right),$$
where “COM” and “rel” refer to the Center of Mass and relative motion, respectively. This system is therefore factorized in the new basis of $\left(r,\;R\right)$, i.e., it is exactly separable into two subsystems: the center of mass and the relative motion, even though the interaction term $v\left(r_{1}-r_{2}\right)=v\left(r\right)$ can be arbitrarily strong.

This illustrates another important point: even in the case when interactions between components are strong, a system may be separable and therefore unconscious.

Furthermore, transformations such as the one used in the example above can be applied to any physical system. In standard quantum mechanics (and therefore in the classical limit of quantum mechanics as well), one can change the Hilbert space basis in which a system is considered to a different basis using unitary transformations [32,47]. A system that is non-separable in one basis may become separable in a different basis, as we have just seen. However, the factorization problem—finding the basis in which the wave function is factorized into the largest number of functions describing independent subsystems—has not been solved in its general form [32].

Since the description of the world in quantum mechanics is equally true under any unitary transformation of the basis, it seems proper to posit that the amount of consciousness should be conserved under these unitary transformations. Therefore, sampling all the unitary transformations is not necessary, and there should be a way to compute the amount of consciousness in any basis. Below, we suggest that a metric of the amount of consciousness utilizing a concept such as the *entanglement entropy*—a measure of the degree of quantum entanglement between subsystems in a composite quantum system (see, e.g., [48])—may be appropriate (see Section 2.9). However, while such a metric can be defined, computing it in practice can be very difficult and may benefit from sampling different unitary transformations of the basis. In this respect, the computation may be similar in difficulty to the case of computing the Phi measure of IIT [6,49,50,51,52,53,54,55].

The conceptual takeaway is that one needs to keep in mind the possibility of finding separable components of the system under various unitary transformations. The combinations of the degrees of freedom that remain non-separable under such transformations constitute the conscious entities, where the amount of consciousness in each such entity is determined by the degree of non-separability and the number of degrees of freedom. For large multi-partite systems, the appropriate degrees of freedom that remain non-separable may very well be some macroscopic variables such as membrane voltages of every neuron in a neuronal ensemble, rather than the underlying microscopic variables such as the positions of every elementary particle that the neuronal ensemble is composed of.

### 2.6. Macroscopic Objects and Biological Systems

The discussion above shows that strong interactions between system’s parts do not necessarily imply non-separability and, hence, consciousness. If that was not the case, our hypothesis would run into difficulties, since we would then expect high consciousness for any macroscopic object—that is, an object consisting of many particles and therefore containing many degrees of freedom—with sufficiently strong interactions between them, even if the system in question is simply a pot of water or a slab of copper.

That is emphatically not the case. Most known strongly interacting systems of particles, such as solids, liquids, molecular complexes, etc., etc., appear to be separable or near-separable into many one- or few-dimensional subsystems. Strongly correlated systems, such as electrons in solids, are probably mostly separable. Elementary particles (quarks, protons, neutrons, electrons, neutrinos, etc.) constituting building blocks of physical systems can usually be separated to a great degree of approximation due to the orders of magnitude difference between the interaction energies within nuclei, atoms, etc., and between them (Figure 3A,B). What is left then in strongly correlated systems are usually a small fraction of overall degrees of freedom, such as “free” electrons or other particles/quasiparticles in solids or liquids, waves, etc. These remaining degrees of freedom may still have strong correlations, but often these strong correlations lead to more separability. Typically, since the remaining degrees of freedom represent identical particles, transformations can be made that reduce the function describing the time evolution of the system state to a product of identical functions with only one or a few degrees of freedom each. This often takes on a form of excitations or quasiparticles (holes, phonons, polaritons, etc.) emerging from the sea of interacting atoms and electrons and behaving close to being independent from each other. The function describing the time evolution of the state of the whole system (e.g., its wave function) is then separable or approximately separable into a large number of functions, each describing the states of a small number of degrees of freedom—such as inner components of atoms, vibrations of the crystalline lattice, flow of electric currents, etc.

One may wonder if more exotic states of matter, such as Bose–Einstein condensates (BECs) or superconductors present more substantial difficulties. However, such systems do also seem to be separable into many small parts, even if the parts are substantially different from the underlying elementary particles. The hallmark of both the BECs and superconductors, for example, is that a large number of particles (electrons, atoms) take on identical states and can be described with just one or a few degrees of freedom (such as the order parameter in the Gross–Pitaevskii equation), plus various quantized excitations [56,57,58,59,60,61]. Even phase transitions, which may be characterized by infinitely long correlations (such as in the case of second-order phase transitions), appear to have the same property: The underlying particles may become strongly correlated across vast distances, but that means that unitary transformations can be found that separate the state function into a product of many low-dimensional functions that follow approximately the same dynamics.

This issue is described in detail by Tegmark [32], and the salient observation is that typically it may be rather difficult to find a highly non-separable system. The most common situation, in fact, appears to be the one where a system is mostly separable due to the symmetries and the hierarchy of interaction energies that differ by orders of magnitude between scales.

Compared to ‘inanimate matter’, biological systems are characterized by more heterogeneity, asymmetry, and mixing of scales. However, even for such systems, a vast amount of the degrees of freedom are likely separable from each other. Most of the subatomic degrees of freedom are relatively independent from larger-scale phenomena (Figure 3A); the movements of atoms themselves can largely be separated from the movements of their various arrangements in molecular moieties such as rings and amino-acids (Figure 3B); and these in turn are relatively independent of the larger-scale dynamics of whole proteins, patches of membrane, segments of DNA, etc. (Figure 3C). At the level of organelles and cells, including neurons, the more microscopic degrees of freedom are also likely separable in many cases (Figure 3D). Indeed, formalisms such as the one employing the Hodgkin-Huxley equations are often sufficient to describe to a high precision the dynamics (such as action potentials) relevant to neuron’s communication with other cells [62,63,64]. Such equations operate with membrane voltages and ionic concentrations, rather than the states of individual ion channels. At the level of whole organs, even brains, dynamics of individual cells may be separable from higher-level degrees of freedom (Figure 3E). For example, states of neuronal ensembles distributed across the brain may be more relevant at that level than the states of every neuron. Beyond this level of organ or perhaps an organism, separability is typically even more widespread, since organisms usually have very low-bandwidth channels of interaction with the rest of the world.

The conclusions that we can draw from these considerations so far, in the light of our hypothesis, are two-fold. First, the vast majority of degrees of freedom in almost any physical system, including biological ones, are separable or near-separable into low-dimensional subsystems that have very little to zero consciousness. Second, the only physical system we know to be conscious—the brain—seems unique in that, despite also being subject to rampant separability of most of its degrees of freedom, it somehow manages to maintain a relatively highly non-separable state with some of the remaining degrees of freedom.

Strictly speaking, the latter point is a supposition, since we do not know for certain how separable or non-separable the brain really is. However, this idea does not seem surprising, given that the brain is often considered to be the most complex piece of matter in the Universe, and its structure and dynamics (especially that of the presumed ‘seat of consciousness’, the thalamo-cortical system [1,2,30,65,66,67]) exhibit a balance between not-too-weak and not-too-strong interactions among many heterogeneous partners, across multiple length- and timescales (nm to cm and ms to years) [68], and without too many obvious symmetries. What the considerations above add, is that the non-separable conscious state in the brain likely forms over a relatively small number of mesoscopic/macroscopic degrees of freedom such as the neuronal membrane voltages and activity states of distributed neural ensembles. The astronomical numbers of all the other degrees of freedom, starting all the way from the elementary particles the brain consists of, are separable into one- or low-dimensional subsystems with little to no consciousness in each subsystem (Figure 3). These considerations are consistent with the IIT’s view on the causal emergence of a conscious complex relying on macroscopic components of the system rather than its microscopic constituents [69,70]. They are also congruent with the idea that brains organize spatio-temporal nestedness that supports consciousness, which is a central concept in the Temporo-spatial theory of consciousness [17,35,71].

It is worth noting that the macroscopic degrees of freedom still may provide for a highly dimensional non-separable system. There are on the order of 10^10^ neurons in the human thalamo-cortical system [72,73], and even assuming that only a fraction of them form the main non-separable state, the dimensionality of that state still may be immense. If the correct level of granularity is neural ensemble, one can still easily expect hundreds or more such ensembles to form non-separable states (for reference, at least 180 areas are identified in the human cortex [74]), still providing for a rather high dimensionality.

Finally, it is useful to keep in mind that separability of degrees of freedom may not occur cleanly along the macroscopic/mesoscopic/microscopic lines, which themselves are imprecise definitions. It is entirely possible that the main conscious non-separable complex in the human brain consists of both macroscopic and microscopic degrees of freedom. For example, the Orch OR mechanism of consciousness is typically assumed to take place in the quantum coherent states of spins in microtubules [11]. Other suggestions implicate spins and long-distance communications between them in the brain via photons [75]. While no direct proof of these suggestions has been obtained yet, they are consistent with our hypothesis, as long as such microscopic degrees of freedom are interacting with each other and perhaps other macroscopic degrees of freedom in a non-separable way.

### 2.7. Feedforward Systems Are Unconscious

An important class of systems in biology and among human-made machines are feedforward systems. In this case, interactions, or “connections”, between the system’s components are asymmetric (e.g., interactions between neurons via chemical synapses) and no component receives connections back from the components that it connects to (no such feedback is present either via direct back-connections or indirectly via a chain of connections with other nodes). This is illustrated in Figure 4. Examples of feedforward systems include sensory inputs to the brain such as the inputs from retina to the thalamus (although some amount of feedback to the retina from the rest of the brain exists [76]) and some of the most successful deep neural networks currently in widespread use [77].

In the framework of IIT, purely feedforward systems have zero consciousness [6]. We now show that the same holds under our hypothesis.

Consider a feedforward system with components (nodes) 1, 2, etc., the states of which are described by variables $x_{1},x_{2},\;\mathrm{etc}.$ A set of nodes, with the states described by variables $inp_{1},inp_{2},\;\mathrm{etc}.,$ provide inputs into the system and do not receive any connections from the rest of the system (Figure 4). For any node i, its state $x_{i}$ is fully determined by the states of all the preceding nodes that connect to it, and the node i has no effect on the preceding variables. This is also true for all the nodes sending connections to node i, and then for all the preceding nodes, and so on until we reach the input nodes. Therefore, the value of $x_{i}$ is fully determined by $inp_{1},inp_{2},\ldots \;$, and we can replace it by some function,
$$x_{i}=g_{i}\left(inp_{1},inp_{2},\ldots ,t\right).$$

In a feedforward system, this is a true function, meaning that nothing in this function depends on $x_{i}$. This is a simple, but fundamental difference from the non-feedforward case, where the states of the components influencing $x_{i}$ in turn depend on the evolution of $x_{i}$ itself.

The same argument can be applied to each node in a feedforward system, and ultimately the state of every single node can be traced via a potentially complicated function to the state of the external inputs to the system. For the function f, describing the time evolution of the full system’s state, we then obtain:$$f\left(x_{1},x_{2},\ldots ,\;x_{i},\ldots ,\;inp_{1},\;inp_{2},\ldots ,t\right)=f\left(x_{1},x_{2},\ldots ,\;g_{i}\left(inp_{1},\;inp_{2},\ldots ,t\right),\;\ldots ,inp_{1},\;inp_{2},\ldots ,t\right)=$$
$$f\left(g_{1}\left(inp_{1},\;inp_{2},\ldots ,t\right),\;g_{2}\left(inp_{1},\;inp_{2},\ldots ,t\right),\ldots ,g_{i}\left(inp_{1},\;inp_{2},\ldots ,t\right),\;\ldots ,inp_{1},\;inp_{2},\ldots ,t\right)$$
$$=F\left(inp_{1},\;inp_{2},\ldots ,t\right),$$
where we replaced each variable by the function of the inputs, $x_{1}=g_{1}\left(inp_{1},\;inp_{2},\ldots ,t\right),\;x_{2}=g_{2}\left(inp_{1},\;inp_{2},\ldots ,t\right),\;$… and used notation F() for the function describing the state of the feedforward system that depends explicitly only on the inputs (and their time evolution).

Does this mean that the function describing the state of a feedforward system is separable? The answer is yes:$$f\left(x_{1},x_{2},\ldots ,\;x_{i},\ldots ,\;inp_{1},\;inp_{2},\ldots ,t\right)=F\left(inp_{1},\;inp_{2},\ldots ,t\right)=F\left(inp_{1},\;inp_{2},\ldots ,t\right)\;I\left(x_{1},t\right)I\left(x_{2},t\right)\ldots I\left(x_{i},t\right)\ldots$$
where $F\left(inp_{1},\;inp_{2},\ldots ,t\right)$ does not depend on any of the system’s degrees of freedom and we introduced $I\left(x,t\right)\equiv 1$.

Thus, the system is fully separable and, according to our hypothesis, has zero consciousness, consistent with the IIT result.

More generally, description of a system has to take into account its environment too. Thus, a feedforward system with some inputs cannot be considered on its own—it is a part of the Universe that contains the system generating inputs as well. (It is possible, however, that this input-generating system together with the feedforward system have zero interactions with the rest of the Universe and are thus separable from it.) In this view, the state function describing the system generating the inputs AND the feedforward system receiving these inputs depends only on the variables of the system generating the inputs—and the feedforward system receiving such inputs is simply an appendix feeding off the dynamical variables of the input-generating system, whereas its own degrees of freedom are fully separable, as shown above.

It is also interesting to note that for any deterministic system its state at any time moment is fully determined by the initial conditions and external inputs over time up to that moment. This may seem like a case that is similar to a feedforward system, but there is a key difference, which ultimately shows in the equations of motion. For a feedforward system, these equations are reduced to functions that explicitly depend on the inputs only, whereas for a non-feedforward system, including deterministic ones, the equations of motion depend on the interactions between the system components. This is then reflected in the state function describing these systems: for a feedforward system, it is fully separable and depends on the inputs only, as shown above, whereas for a non-feedforward system it will depend on the internal degrees of freedom and may not be exactly separable. For non-deterministic feedforward systems, the considerations above still hold, with functions $g_{1}\left(inp_{1},\;inp_{2},\ldots ,t\right),\;g_{2}\left(inp_{1},\;inp_{2},\ldots ,t\right),\;$… becoming stochastic rather than deterministic, but still remaining dependent on the inputs only, which results in full separability.

These considerations show that, even though the dynamics of the system’s components may be complex, what really matters are the internal interactions. This is consistent with the examples considered by Tononi and colleagues in the IIT framework [5,6]: a feedforward system can generate exactly the same output as a system with feedback, but the amount of consciousness, Phi, of this feedforward system is still zero, which is not the case for the system with feedback.

### 2.8. Non-Separability and Entanglement in Quantum and Classical Systems

The concept of non-separability is intertwined with another, perhaps more often discussed concept, that of quantum entanglement. We now consider the similarity and distinction between these two concepts and discuss which formalism may offer a convenient framework to study non-separability in physical systems, especially in cases where the classical limit of quantum mechanics is involved.

Quantum entanglement has been much discussed as a potential mechanism for consciousness (e.g., [11,75,78]). However, brains are ‘warm’ and ‘wet’, meaning that degrees of freedom in the brain are subject to substantial influence from the thermal motion. This implies that a quantum coherent state is unlikely to survive for long enough intervals of time to support consciousness, which we know to operate on the timescales of ~50–100 ms and beyond [36,37,38]. Proposals that quantum entanglement underlies consciousness usually posit that such thermal influences may be overcome in certain degrees of freedom, such as nuclear or electron spins in proteins, e.g., in microtubules. While this may be possible, the degree to which this is realized in the brain and supports consciousness is unclear. So far, the only phenomenon that has been relatively convincingly shown to require strictly quantum processes while also having a functional role in the brain (specifically in the retina) is the animal magnetoreception via the radical pairs mechanism [79,80,81,82,83], though even that is still a matter of some debate. Chemical reactions, which inherently rely on quantum processes, occur throughout the brain, but they mostly seem to be coupled to the thermal bath, which prevents coherence and entanglement—with the radical pairs mechanism being one relatively well-characterized exception. Thus, the extent to which quantum entanglement may occur in the brain and may be involved in consciousness remains to be elucidated. (One interesting avenue for doing so is via psychological experiments on violation of the Bell-type inequalities for cognitive systems [84,85,86]).

In quantum mechanics, the concept of entanglement includes at least two separate notions—non-locality and non-separability. Non-locality means that, in the entangled pair of particles, knowledge of the state of one particle immediately tells us the state of the other, even if this other particle is far away. And non-separability is what we have been discussing so far, that the wave function describing such two particles cannot be factorized into the product of two functions, each describing one of the particles. This is an important distinction: quantum nonlocality is different from non-factorizability of the state vector. Both are features of quantum entanglement. Among the two, non-locality is a purely quantum phenomenon, whereas non-separability can exist in classical systems [87,88]. Indeed, electromagnetic fields in classical regime have been shown to exhibit “classical entanglement” (see, e.g., [87,88,89,90,91,92]), which, however, should be more properly called classical non-separability. Furthermore, classical composite (macroscopic) systems may demonstrate correlations similar in appearance to those due to quantum entanglement if the macroscopic states are considered at a time scale that permits microscopic interactions within each macroscopic time step [69,93].

Currently we do not know whether consciousness arises from purely classical processes, thus involving non-separability but not non-locality, or depends on quantum phenomena too, in which case it may involve non-locality (and therefore quantum entanglement). Is there a physical formalism that could possibly account for both options and also describe the possible transition between the two? Of course, all systems are described by quantum mechanics, including those in the classical limit. However, the typical quantum formalism employing the wave function is often hard to use to describe classical systems. One equivalent approach involves the Wigner function [40], which depends not on the positions only or the momenta only, as the wave function may, but on both positions and momenta of particles in the system (i.e., it is defined in the phase space). It is connected to the wave function by the following transformation:$$W\left(x,\;p,t\right)=\frac{1}{2\pi \hbar }\int e^{-\frac{ipy}{\hbar }}\psi \left(x+y/2,t\right)\psi^{*}\left(x-y/2,t\right)dy,$$
where $\psi \left(x,t\right)$ is the wave function defined in the space of positions, and the positions and momenta are multi-dimensional vectors representing all the particles in the system.

In contrast to the wave function, which is complex, the Wigner function has real values. However, unlike a true probability density (such as the modulus squared of the wave function), the Wigner function can have negative values, as well as non-negative ones. Most interesting for us here is the classical limit, and obtaining it may often be more tractable for the Wigner function than for the wave function. It is still not as straightforward as setting the limit to zero for the Planck constant, ℏ→0 (in general, obtaining the classical limit for quantum systems may involve additional conditions, such as accounting for the mean-field interactions: see, e.g., [94,95,96] and references therein). However, when performed carefully, it can be shown (see, e.g., [41,97]) that in the classical limit the Wigner function becomes the Liouville density function in phase space. Liouville density function is a true probability density in phase space (all negative values in the Wigner function are eliminated in the classical limit), representing the probabilistic ensemble of the system’s states.

A detailed discussion of the properties of the Wigner function is beyond the scope of this paper and can be found elsewhere (see, e.g., the broadly accessible treatment in [41]). Of relevance are the following few observations: the Wigner function fully describes the state of a system, similar to the wave function; there is a direct correspondence between the Wigner function in the general quantum case and the Liouville density function in the classical case; if the wave function is separable, the Wigner function is also separable. Thus, in all the preceding material of this paper, the functions describing the state of the system can be Wigner functions (where each degree of freedom would be described by a pair of values such as position and momentum, rather than just one value), just as they can be wave functions. The treatment of non-separability applies in the same way. Furthermore, for classical systems, all the same considerations will apply to the Liouville density function. Classical systems can be non-separable, which is described as the case when the Liouville density function of multiple degrees of freedom cannot be represented as a product of functions depending on the subsets of these degrees of freedom.

Two main conclusions follow. First, the Wigner function/Liouville density function formalism appears to offer a convenient framework for describing non-separability—and, if our hypothesis is correct, consciousness as well—for both quantum and classical systems. Second, because the Liouville density function describes a *probabilistic ensemble* in the phase space rather than an individual trajectory, the proper description of non-separability and consciousness *relies on the properties of the ensemble* and not a single realization of it, even in a purely classical case. This unintuitive result is consistent with the IIT, where the calculation of the amount of consciousness, Phi, considers not only the actual past, current, and future states of the system at a given time step, but also *what those states could possibly be* [5,6].

Notably, Wigner function formalism has been instrumental in establishing separability criteria for physical systems. In particular, it was used to show that in the case of continuous degrees of freedom the so-called Peres-Horodecki criterion provides uncertainty principles that must be obeyed by all quantum separable states, whereas for all bipartite Gaussian states, the Peres-Horodecki criterion is a necessary and sufficient condition of separability [98]. A related criterion for entanglement of a two-part quantum system has been established [99]. Remarkably, Diaz et al. [100] described a *classical* analog of the quantum covariance matrix and, in the case of Gaussian states, derived expressions for classical analogs of the purity, linear quantum entropy, and von Neumann entropy for classical integrable systems.

As an example, Diaz et al. [100] considered classical coupled harmonic oscillators, described by the Hamiltonian
$$H\left(q_{1},q_{2},p_{1},p_{2}\right)=\frac{1}{2}\left(p_{1}^{2}+p_{2}^{2}+Aq_{1}^{2}+Bq_{2}^{2}+Cq_{1}q_{2}\right).$$

Quantum purity, $\mu \left(\hat{\rho }\right)=Tr\left({\hat{\rho }}^{2}\right)$, where $\hat{\rho }$ is the density operator, equals 1 for pure states, and for mixed states $0<\mu \left(\hat{\rho }\right)<1$. They showed that the classical analog of purity, ${\tilde{\mu }}^{cl}$, for the individual components 1 and 2 of the coupled oscillators above is given by the expression
$${\tilde{\mu }}^{cl}\left(1\right)={\tilde{\mu }}^{cl}\;\left(2\right)=\sqrt{\frac{4AB-C^{2}}{4AB}}$$
whereas for the compound system including both components 1 and 2 it is ${\tilde{\mu }}^{cl}\left(1,\;2\right)=1$. Thus, the combined system {1, 2} is pure, whereas subsystems {1} and {2} are generally not, that is, they are not separable from each other. The system does become separable in the special case of C=0 since in that case ${\tilde{\mu }}^{cl}\left(1\right)={\tilde{\mu }}^{cl}\;\left(2\right)=1$, which is the expected result for uncoupled oscillators.

Thus, the Wigner function formalism has been already highly influential in studying separability and non-separability in both *quantum and classical* cases and, in light of our hypothesis, is likely to be useful in theoretical studies of consciousness.

### 2.9. Measure of Consciousness

Can our hypothesis of equivalence between non-separability and consciousness inform the choice of measures for quantifying the amount of consciousness in a system? Below, we offer some general considerations regarding this question.

Let us first consider neurons in the brain and assume for the sake of a simple example that all phenomena relevant for computing the amount of consciousness occur in the classical limit. The previous section showed that one needs to estimate the properties of the statistical ensemble and not just a single classical realization of the system with all degrees of freedom. Assuming also that most of microscopic degrees of freedom are separable, perhaps one may limit the consideration to only the macroscopic degrees of freedom such as the membrane voltage or firing rate of each neuron. Then we need to describe the separability properties of the Liouville density function for such firing rates and their conjugate variables such as the time derivative of the firing rates (to construct the complete phase space).

Interestingly, we can see that the results might be different depending on the timescale considered. E.g., for microsecond time scales, the neurons effectively do not interact with each other, especially across the whole brain, since the axonal propagation speed and synaptic delays set the time of the downstream response after upstream spike to a few milliseconds. At these time scales, the system is likely very much separable. But on large timescales, such as ~50–100 ms that is often considered the time grain of human consciousness [36,37,38], most neurons will have enough time to communicate, and the system may become non-separable. We may then speculate that non-separability could possibly be estimated, at least for binary partitions, by assuming ergodicity over the time scale of interest (such as 100 ms). One can accumulate instantaneous values of the firing rates and their derivatives from smaller time bins such as 10 ms to obtain the distribution during the longer interval of 100 ms. This distribution could possibly be used as a proxy to the Liouville function, and one could attempt to test whether it is separable or not along bipartitions across the degrees of freedom.

Unfortunately, such an approach can be complicated by the following issues. Consider two neurons, with firing rates x and y. Ignoring their time derivatives for simplicity, the procedure outlined above will furnish an estimate of the distribution of x and y, and one may want to check the correlation between x and y as a means to test for separability. Indeed, one can show easily that if the function describing the state of the system (such as the density function) is separable over x and y, then the correlation between them is zero (see example in Figure 5 on the left). Thus, non-separable state is required to obtain non-zero correlations. Non-zero correlations are often observed for neurons in the brain, and one may take that as an indication of non-separability. However, the situation is more complicated even for the case of just two variables. For example, a non-separable function such as $f\left(x,y\right)=exp\left[-{\left(x+y-A\right)}^{2}/\left(2\;B^{2}\right)\right]$ leads to non-zero correlations between x and y (Figure 5, middle). But, as we discussed before, we should consider whether the degrees of freedom can be separable in a different basis. Indeed, here we can simply transform the coordinates by rotating them 45 degrees, in which case the density function describing the system becomes separable (it only depends on one of the two coordinates after the transform), and the correlation between the new coordinates is again zero (Figure 5, right). While this does not mean that ergodic estimates of the Liouville function outlined above will be useless, this simple example underscores again the difficulty of practical computation of non-separability.

Is there a solution to this problem? It may be informative to turn to studies of the related problem of entanglement in quantum mechanics, where many measures are in use. One standard measure is the entanglement entropy, which applies to pure quantum-mechanical states and can be defined, e.g., for a bipartition of system as the Von Neumann entropy of the state of one or the other resulting subsystems (the result is equal for either of the two subsystems). This quantity is zero if the system’s state is non-entangled (separable) and positive otherwise. Additional measures have been introduced to quantify the extent of entanglement for mixed states, such as the relative entropy of entanglement, logarithmic negativity, and others.

While active research continues in this area, approaches to detect entanglement or compute the degree to which a system is entangled (typically, for bipartite systems) have been developed for a long time (see, e.g., refs. [48,98,99,100,101]). It is thus interesting to consider the possibility of applying similar metrics for the non-separability question, whether in quantum or classical case. Intriguingly, it has been suggested that in a quantum system having a well-defined classical counterpart, quantum correlations may be described by the mutual information of the corresponding classical system [102]. Computations of entanglement entropy starting from Wigner function have been achieved [103,104,105], and may be used to describe the degree of non-separability in a classical system, taking the transition from the Wigner function to the classical Liouville density function in the phase space. As discussed above, we would expect in this case that quantum non-locality will vanish, but some degree of non-separability may remain [100]. The entropy-based measures characterizing non-separability, such as entanglement entropy, will then have non-zero values, although one would expect those values to be lower than for a similar quantum system in an entangled state containing non-locality.

Thus, we may suggest that entropy-based measures such as the entanglement entropy, especially when computed using the Wigner function or its classical equivalent the Liouville density function, can be useful in characterizing the amount of consciousness in a physical system. Beyond characterizing non-separability of bi-partite systems, such as can be established using the Von Neumann entropy, it is important to consider compositions other than bi-partite. In this case, logarithmic negativity can be a useful measure [106]. For example, it was shown that the entanglement properties of the ground and thermal states of a closed chain of harmonic oscillators coupled by translationally invariant Hamiltonian can be completely characterized analytically using logarithmic negativity [107]. However, proposals of specific formulas and characterization of their properties will require substantial future work, as the measures of entanglement discussed above are typically very difficult (NP-hard) to compute.

It will also be interesting to consider how such measures may relate to the computation of the Integrated Information Phi in the IIT [5,6,49,50,51,52,53,54,55], which is notoriously difficult to compute as well. Qualitatively, it seems plausible that Phi and entropy-based measures of entanglement have much in common, as both characterize the degree to which the system cannot be reduced to a simple sum of its parts. Indeed, a key concept in the IIT is “*integration*”, meaning that in a conscious state the constituent components are integrated in a unified ‘whole’. Our hypothesis implies integration as well, in the sense of non-separability and not necessarily information processing. Importantly, Tononi and colleagues themselves warn against understanding the “information” in IIT as the Shannon information. Furthermore, as Tegmark showed [32], integrated information in Shannon’s sense tends to be very small—especially so in quantum systems, where the maximum of integrated information one may be able to find is ~0.25 bit. Again, this underscores the point that integrated “information” in IIT should not be taken as Shannon information, but rather as the ability of components of the system to influence each other causally.

Another aspect of consciousness according to IIT is “*differentiation*”—the property describing the observation that a conscious state may contain multiple attributes at once [5,6]. For example, if one has an experience of seeing a landscape, that may at once include seeing a mountain, trees on the mountain, the sky above, etc., etc. A related interpretation is that of a temporal differentiation [5,6,108,109,110,111], in that a rich experience such as watching an engaging movie contains multiple percepts in unit time (say, within 10 s), whereas a less rich experience such as watching TV noise contains few percepts within the same unit time (while pixels may change dynamically in a TV noise, the experience to a human remains just that of a boring, undifferentiated TV noise). While integration is clearly a part of our hypothesized framework of non-separability underlying consciousness, it is less clear that differentiation necessarily belongs to it as well. From first-hand experience, rich or more limited content of consciousness, such as watching an engaging movie vs. watching a dark sky, does not feel fundamentally different. Both are perfectly conscious experiences; however, one is full of informational content and the other is quiet or relatively “empty”.

We may therefore suggest that what matters for the amount of consciousness is the number of non-separable degrees of freedom rather than how fast or slow the dynamics along these degrees of freedom happens, or how rich that dynamics is. In this view, a number quantifying the amount of consciousness would describe the system’s capacity to have experiences (i.e., how it feels to have certain kind of experience, such as how it feels to experience space [112,113]), rather than the immediate richness of the content of the perception (i.e., what exactly is being experienced, such as seeing a certain object in space). This seems consistent with the idea expressed above, that the amount of consciousness is determined by the whole statistical ensemble, rather than a single trajectory (perhaps the latter may underlie the immediate differentiation of the percept). It also should be noted that this view may not be inconsistent with the way Phi is computed in the IIT, as the integration is clearly a part of calculation of the Phi, whereas the differentiation is less obviously so.

To conclude, entropy-based measures such as the entanglement entropy appear to be natural candidates for quantifying the amount of consciousness in the framework of our hypothesis. But we must also keep in mind that a single measure may not be sufficient to characterize all important aspects of consciousness. It will therefore be advisable to research additional measures that could characterize various different aspects of conscious states, such as the differentiation described above, or perhaps the effective dimensionality of the state [114,115,116], to name a few.

## 3. Discussion

We presented the hypothesis that non-separability is equivalent to consciousness, in the sense that non-separable degrees of freedom form a unified ‘whole’, a complex that exists intrinsically for itself in the multi-dimensional space defined by these degrees of freedom. We posit that such unified multi-dimensional existence constitutes consciousness. This view naturally suggests that the non-separable degrees of freedom are ‘in’ and the rest are ‘out’ of this complex, accounting for both integration and exclusion principles of the IIT [5]. Thus, any system can be represented as a collection of big and small complexes—some containing only a single degree of freedom, separable from all the others and thus completely unconscious, and others containing two or more non-separable degrees of freedom and therefore having some amount of consciousness. The amount of consciousness is determined by the number of non-separable degrees of freedom in the complex and the extent of their non-separability. Presumably, most such complexes that form in various physical systems contain very small amounts of consciousness that would feel like nothing from a human point of view, but some systems such as brains manage to produce particularly high-dimensional and strongly non-separable complexes resulting in levels of consciousness we are familiar with.

We saw that non-interacting systems are exactly separable and therefore unconscious. We also showed that feedforward systems are exactly separable and therefore completely unconscious, an observation that is consistent with predictions of the IIT. Importantly, interacting systems can also be exactly separable, i.e., unconscious. In some cases, a system can be completely separable despite arbitrarily strong interactions between the system’s components. This underscores the notion that non-separability may not be as widespread or easily achievable as may appear from the first glance [32].

It, therefore, appears that many complex systems with heterogeneous interactions are also mostly separable, including familiar physical systems such as liquids, gases, or metals, more exotic states of matter such as superconductors, and even biological systems such as biomolecules, cells, and organs. It seems likely that all such systems are separable into an astronomical number of tiny complexes, each consisting of one or a few degrees of freedom (e.g., separate particles or their collective modes of motion, such as rotations of molecular groups or quasi-particles such as holes or excitons) and only loosely influencing each other. Again, brains are probably unique in creating much larger and sophisticated non-separable complexes. However, we speculate that even in this case, the largest complexes in the brain, which presumably support what we experience as human consciousness, are separable from the vast majority of the brain’s degrees of freedom: subatomic, atomic, molecular, subcellular, and cellular. These largest complexes likely consist of only high-level collective degrees of freedom such as neuronal membrane voltages or population firing rates, therefore utilizing only a tiny fraction of the brain’s total degrees of freedom.

We also discussed quantum entanglement, which has two aspects—non-locality and non-separability. In our hypothesis, non-locality per se is not necessary for consciousness, whereas non-separability is. Therefore, in the framework presented here, consciousness does not require quantum effects (including entanglement) and may occur in the classical limit. The hypothesis, of course, does not prescribe whether consciousness is a purely quantum or a purely classical phenomenon, and in principle permits the existence of both classical and quantum systems that are conscious. We argued that, in the light of this observation, the Wigner function-based description of quantum mechanics and its equivalent in the classical case, the Liouville density function-based description, appear to be well suited for exploring non-separability and, thus, consciousness, in both quantum and classical systems.

It is important to note that, although our hypothesis does not preclude consciousness in classical systems, in the quantum case it is consistent with the Orch OR theory [11]. More specifically, the Orch OR framework and that presented here describe different facets of consciousness. Orch OR describes how the moments of consciousness occur as a result of the classical realization of a specific state among multiple states that a quantum system may occupy (i.e., when quantum superposition is ‘reduced’ to classical certainty). Our hypothesis adds the conceptual framework describing how much consciousness there is in the system, which is determined by the non-separability of the system’s wave function (or, equivalently, its Wigner function). The Orch OR principle is outside of the ‘standard’ quantum mechanics and is related to the so-called ‘interpretations’ of quantum mechanics [117,118,119], which explain how a probabilistic quantum superposition may be related to the classical single-state certainty of individual state realizations. By contrast, the non-separability hypothesis operates entirely within standard quantum mechanics, as it explicitly involves non-separability of the state vector, which is an inherently probabilistic description of the ensemble of states that a system may occupy, even in the classical case (as in the Liouville density function formalism). Thus, the non-separability hypothesis and Orch OR theory provide complementary principles, reconciling the standard quantum mechanics and ‘outside of quantum mechanics’ views and describing different aspects of consciousness: how much consciousness there is and when and how the moments of consciousness occur.

On the other hand, we emphasized throughout the paper that our hypothesis is conceptually similar to the IIT. We started from the IIT “axioms” or features of consciousness, which reflect the most basic observations of the nature of consciousness, and discussed how these features, especially the integration and exclusion, may be realized if one assumes that non-separability is equivalent to consciousness. Throughout the paper, we saw that this hypothesis leads to some of the same conclusions as in IIT. These include the observations that feedforward systems are unconscious and that the amount of consciousness is determined not simply by the specific classical realization of the system’s state, but by a statistical description of *possible* states that a system may occupy. These commonalities suggest that our hypothesis and IIT may in fact be equivalent. Attempting a rigorous mathematical proof of such equivalence will be an interesting future direction of research, although formally connecting the information-theoretic IIT framework with the Hamiltonian- and wave-function-based approach of the non-separability hypothesis will clearly be a difficult undertaking. It is worth noting that IIT is typically formulated for discrete systems, whereas the framework proposed here applies to systems that may contain both continuous and discrete degrees of freedom.

It also remains to be seen whether the metrics of the amount of consciousness that we discussed, such as entropy-based measures of entanglement, are consistent or equivalent to the integrated information measure of IIT, Phi. Conceptually, they appear similar (and similarly very hard to compute for more than a few degrees of freedom), but details of the computation may matter in determining the relationship between such metrics. We also argued that one may need to use multiple metrics to characterize consciousness, such as entropy-based metrics to establish the extent of non-separability in the conscious complex, dimensionality metrics to characterize the inner space supported by the complex, differentiation metrics [5,6,108,109,110,111,120] to describe the contents of the consciousness, etc. Interestingly, both the non-separability hypothesis and IIT offer a plausible foundation for the proposed practical measures of consciousness such as the Perturbational Complexity Index (PCI) [121,122,123,124,125], since the latter depends on the degree of integration between the systems’ components. However, a strict mathematical derivation of PCI from either IIT or non-separability hypothesis (or both) remains to be obtained.

Finally, an important question is whether non-separability as a foundation of consciousness has anything to do with the computational power and efficiency of the underlying system. If we are considering a quantum system, such as in the Orch OR framework, the answer is relatively clear. Quantum entanglement, which includes non-separability, supports unprecedented (for a classical computer) capabilities of a quantum computer [126,127]. Even in the absence of entanglement (i.e., non-locality), a quantum system still can offer computational power and efficiency well beyond a classical counterpart [128,129,130,131], e.g., due to the superposition phenomenon. To be clear, a classical computer can in principle simulate superposition or even entanglement, but will require an enormous amount of time and power to achieve that. In a classical limit, the situation may be less clear, but there are indications that even then non-separable systems offer substantial advantages over separable ones [132,133,134,135,136,137], e.g., in terms of computational power, time required to carry out computations, and efficiency of learning. Thus, it appears that, even in a classical case, a non-separable system realizes a seamlessly integrated many-dimensional ‘whole’, in which complex computations can occur in a more straightforward way than in a separable system.

Understanding the relationship between non-separability and computation will require much more research, but basic considerations above suggest that non-separability is likely beneficial for computation in classical and most definitely in quantum systems. Then it is easy to see that non-separability has an evolutionary advantage. It permits more efficient and faster computations that would improve the organism’s reaction to the conditions in its environment. The equivalence between non-separability and consciousness then means that consciousness has an evolutionary advantage, which offers a fascinating explanation for the evolution of highly conscious animals such as humans and their associated intelligence. Interestingly, computational experiments with simple ‘animats’ showed increase in the amount of consciousness, measured as the IIT’s integrated information Phi, with animat evolution [138,139].

In summary, we hypothesized that non-separability is equivalent to consciousness and that the number of degrees of freedom in a non-separable system and the extent to which they are non-separable determine the amount of consciousness they possess. We saw a number of consequences from this hypothesis, and it will be interesting to investigate many other related questions, such as the following.

What is the exact mathematical relation between this hypothesis, IIT, and Orch OR?What are the appropriate metrics for the amount of consciousness, based on the measures of entropy, dimensionality, differentiation, etc.?How can we test this hypothesis experimentally, given that quantifying non-separability in general is rather difficult?What are the ramifications of the choice of basis for representing a physical system, given that the factorization problem has not been solved in a general case?What are the computational properties of non-separability in quantum and classical cases, how are they realized in the brain, and what evolutionary advantages do they confer?Does biological consciousness in the brain occur as purely classical non-separability, as purely quantum non-separability (and, possibly, entanglement that also includes non-locality), or a combination of the two?

At present, the hope is that this hypothesis can stimulate further work in the community and possibly reconcile the IIT and Orch OR theories in order to unify our understanding of how consciousness emerges from fundamental physical phenomena.

## Figures and Tables

**Figure 1 entropy-24-01539-f001:**
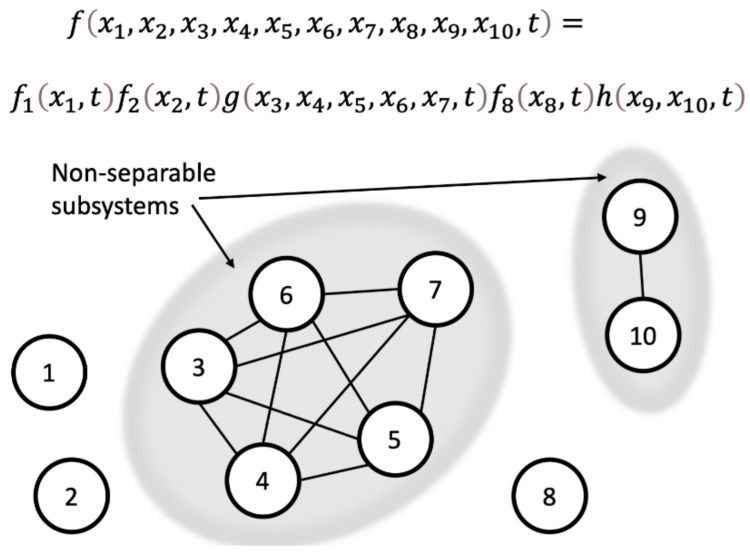
The non-separability concept. Shown is an example of a system with 10 degrees of freedom. The system is described by the function f, which can be decomposed into a product of the functions describing the subsystems: the three subsystems with one degree of freedom each ($x_{1}$, $x_{2}$, and $x_{8}$), one with two non-separable degrees of freedom ($x_{9},x_{10}$), and one with five non-separable degrees of freedom ($x_{3},x_{4},x_{5},x_{6},\;x_{7}$). Interactions between the degrees of freedom are schematically shown as edges. We hypothesize that the extent of non-separability and the number of degrees of freedom involved determine the amount of consciousness. In this example, the non-separable subsystems $x_{9},x_{10}$ and $x_{3},x_{4},x_{5},x_{6},\;x_{7}$ form separate entities, each of which has some amount of consciousness (very small, given the number of degrees of freedom involved).

**Figure 2 entropy-24-01539-f002:**
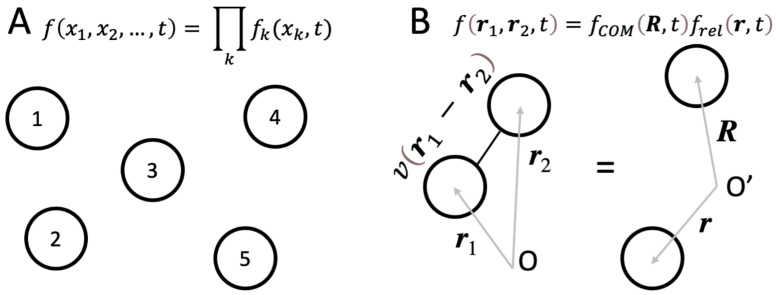
Separable non-interacting and interacting systems. (**A**) A system with components that do not interact with each other. This system is exactly separable, and the amount of consciousness it contains is zero. (**B**) A system of two interacting particles with position vectors $r_{1},\;r_{2}$ (relative to origin O in a coordinate system). The interaction is marked by a black line. By transforming to a different coordinate system (origin marked as O’), where the center-of-mass (“COM”, R) and relative (“rel”, r) motions are considered, one finds that the system is exactly separable (as there is zero interaction between R and r). Thus, it also has zero consciousness, even though the interaction between $r_{1}$ and $r_{2}$ can be arbitrarily strong.

**Figure 3 entropy-24-01539-f003:**
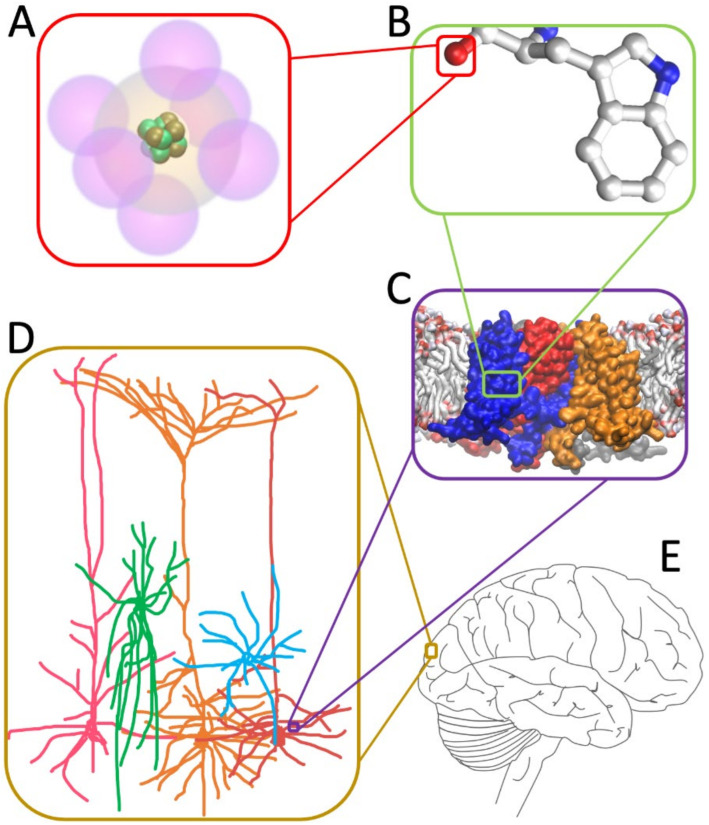
Separability in macroscopic objects such as biological systems. (**A**) Subatomic particles such as protons, neutrons, and electrons interact stably and with high energy within the atom. Most of their degrees of freedom are effectively separable when considering interactions between atoms, where typically the whole atom can be seen as a single particle, perhaps with additional contributions from “outer” electron shells. The nucleus and the electron orbitals are not shown to scale (the real size difference is about 10^5^). (**B**) Molecular moieties often have relatively rigid structure and multiple symmetries. Many degrees of freedom from the constituent atoms may be separable from the major molecular motions, e.g., amino acid side chains bend and rotate while maintaining their overall internal structure. (**C**) In macromolecular complexes immersed in liquids and biological membranes, movements of the numerous constituent molecular residues (e.g., amino acids, lipids, and water molecules) can often be separable from the large-scale dynamics, where many of the relevant degrees of freedom (though not all) may be the rigid-body motions, twists, compressions, and other modes of the macromolecular motions, as well as order parameters of the membrane continuum and concentrations of solutes. (**D**) At the cellular level, such as when considering neuronal activity, many of the underlying macromolecular phenomena may be separable. Higher-level composite degrees of freedom such as membrane voltage and ionic concentration gradients are in many cases sufficient to describe dynamics at this level. (**E**) For whole brains, it is likely that details of the activity within individual cells are separable from the larger scale dynamics of the neurons (reflected, e.g., in the somatic output membrane voltage and action potentials) and neuronal ensembles.

**Figure 4 entropy-24-01539-f004:**
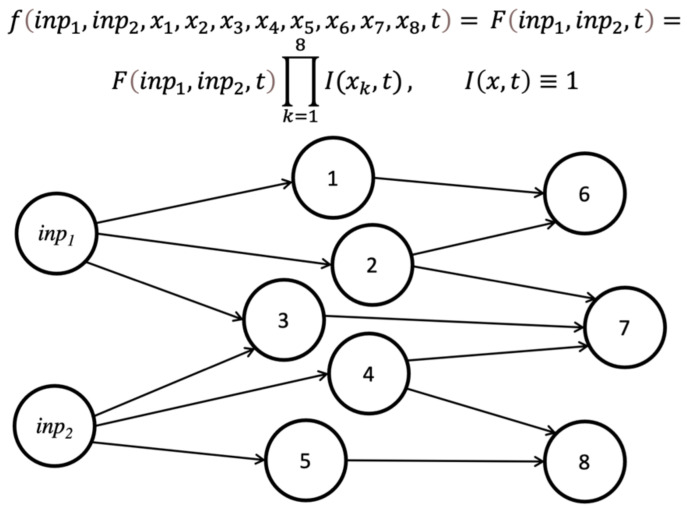
Feedforward systems are unconscious. The system depicted here consists of nodes that interact with each other asymmetrically (as denoted by directed connections). Two nodes, inp_1_ and inp_2_, serve as inputs to the whole system. All interactions are purely feedforward (i.e., a node does not receive connections—either directly or indirectly via a chain of other nodes—back from those nodes that it connects to). Then, the state of every node x_1_, …, x_8_ is determined entirely by the states of the two input nodes, as defined by the function F().

**Figure 5 entropy-24-01539-f005:**
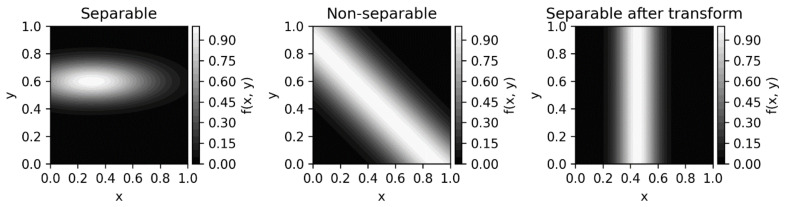
Correlations in separable and non-separable systems. On the left, an example of a system $\left(x,\;y\right)$ described by the separable function $f\left(x,y\right)=f_{1}\left(x\right)f_{2}\left(y\right)$ is shown, where $f_{1}$ and $f_{2}$ are Gaussians. Correlation between x and y is zero. A non-separable example is shown in the middle, using the function $f\left(x,y\right)=exp\left[-{\left(x+y-A\right)}^{2}/\left(2\;B^{2}\right)\right]$. There is a negative correlation between x and y. However, as shown on the right, after coordinate transform (rotation of x and y by 45 degrees), this system becomes separable, with zero correlation between the two rotated degrees of freedom.

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
