# Peer review of "Non-Separability of Physical Systems as a Foundation of Consciousness"

_entropy, 2022, doi:10.3390/e24111539_

Round 1

Reviewer 1 Report

This paper is about the separability as the need for consciousness. Systems with high separability have no consciousness while systems like the brain with high non—separability have high consciousness. The authors discuss current findings, the physical background of their hypothesis and its relevance for IIT and other theories. This is an interesting albeit somewhat speculative paper which raises some questions.

-          What do they refer to when they speak of consciousness? Different dimensions like level, form and content? Or something else?

-          The same goes for non-separability..what is meant by that? Nothing is said in the introduction..

-          The axioms in IIT have raised some critici..see a paper by Bayne in Nueroscience of consciousness…and they are not relaly axioms in the mathematical sense…

-          All the discussion of non-separability focuses almost exclusively around the IIT..ideally I would like to have some independent sources ….

-          More references would be helpful in many parts beyond the IIT realm…..like in the biological systems and unconscious systems…

-          I never hard about the concept of entanglement entropy..may be I missed something? If not some explanation and formalization would be helpful…

-          They distinguish neural correlates and basic physical principles, as I understand..this converges well with the idea of the neural predisposition of consciousness (NPC) as introduced by the group around Northoff (Northoff 2013, Northoff and Heiss 2015m, Northoff and Huang 2017, Northoff and Zilio 2022)…that should ben discussed here

-          Some overview papers of consciousness in neuroscience should be refrred to and cited: Seth and Bayne 2022, Northoff and Lamme 2020…

-          I was wondering how their hypothesis of non-separability relates to time and especially space. The Temporo-spatial theory of consciousness conceives time-space to be key for consciousness: as I understand their mechanisms like spatial nestedness converges well with the here suggested non-separability; see Northoff & Huang 2017, Northoff & Zilio 2022a and b.

-          What is meant by ‘amount of consciousness’? how to measure? How to quantify?

Author Response

I thank the esteemed reviewer for the detailed comments and insightful questions. In the revised manuscript, I addressed all the comments and included the references mentioned by the reviewer. Please see the detailed answers to each point below.

  • “What do they refer to when they speak of consciousness? Different dimensions like level, form and content? Or something else?”

This is mentioned briefly at the beginning of Introduction. I have now added the following paragraph in Sec. 1 of Results to make it clearer.

“Here we use the definition of consciousness as the phenomenon of having a subjective experience [1,2], such as seeing something, having a thought, or experiencing an emotion. Obviously, human conscious experiences are often highly complex, combining perception of various sensory modalities and inner states at the same time and occurring at multiple spatio-temporal levels. Our task here is not to uncover the nature of all the possible aspects of such experiences, but rather to understand how the basic phenomenon of consciousness – the first-hand experience, or feeling anything at all – comes about.”

  • “The same goes for non-separability..what is meant by that? Nothing is said in the introduction..”

I added the following passage to Introduction (the second to last paragraph).

“Mathematically, non-separability means that the function describing the state of a system cannot be represented as a product of functions describing the system’s parts. E.g., if the wave function describing a system of two degrees of freedom, , can be written as the product of the wave functions describing each degree of freedom, , then this system is separable, and if it cannot (), then it is non-separable.”

  • “The axioms in IIT have raised some critici..see a paper by Bayne in Nueroscience of consciousness…and they are not relaly axioms in the mathematical sense…”

Good point. In the revised manuscript, I avoided using the word “axiom” most of the times, and revised the relevant sentence in Sec. 1 of Results to read as follows:

“Such attributes have been enumerated as five features of consciousness  forming the foundation of IIT [5] (originally called “axioms”, although objections have been raised to calling them such [39]),  and we will repeat them here.”

  • “All the discussion of non-separability focuses almost exclusively around the IIT..ideally I would like to have some independent sources ….”
  • “More references would be helpful in many parts beyond the IIT realm…..like in the biological systems and unconscious systems…”

Since this manuscript is intended for the collection on IIT, I thought it was reasonable to focus much of discussion on IIT and refer to the relevant papers. However, I do also cite a number of papers on the Orch OR theory as well as other theories. I also now cite all the papers suggested by the reviewer.

  • “I never hard about the concept of entanglement entropy..may be I missed something? If not some explanation and formalization would be helpful…”

Yes, this should be explained better. The detailed explanation and discussion are in Sec. 9, and I now added a clarification in the sentence where entanglement entropy is mentioned first:

“Below we suggest that a metric of the amount of consciousness utilizing the concept like the entanglement entropy – a measure of the degree of quantum entanglement between subsystems in a composite quantum system (see, e.g., [48]) – may be appropriate (see Sec. 9).”

  • “They distinguish neural correlates and basic physical principles, as I understand..this converges well with the idea of the neural predisposition of consciousness (NPC) as introduced by the group around Northoff (Northoff 2013, Northoff and Heiss 2015m, Northoff and Huang 2017, Northoff and Zilio 2022)…that should ben discussed here”

  • “Some overview papers of consciousness in neuroscience should be refrred to and cited: Seth and Bayne 2022, Northoff and Lamme 2020…”

These are great suggestions. Thank you for highlighting this, so I can fix the unfortunate omission of this work from the previous version of the manuscript. This has now been done in the revised manuscript (see, in particular, the 3rd paragraph of Introduction), and all the suggested references were cited.

  • I was wondering how their hypothesis of non-separability relates to time and especially space. The Temporo-spatial theory of consciousness conceives time-space to be key for consciousness: as I understand their mechanisms like spatial nestedness converges well with the here suggested non-separability; see Northoff & Huang 2017, Northoff & Zilio 2022a and b.

Agreed. I added the following sentence in Sec. 6 of Results.

“They are also congruent with the idea that brains organize spatio-temporal nestedness that supports consciousness, which is a central concept in the Temporo-spatial theory of consciousness [17,35,71].”

  • What is meant by ‘amount of consciousness’? how to measure? How to quantify?

Thank you, it is indeed worth clarifying. I added the following sentences in the last paragraph of Introduction.

“We propose that such non-separable complexes are conscious, and the extent to which they are conscious is determined by the extent of non-separability and the number of degrees of freedom involved. This determines the amount of consciousness each complex has. By amount of consciousness, we mean the measure of consciousness, a hypothetical number that characterizes consciousness and possibly its richness, such as Phi in IIT. We discuss this in more detail in Sec. 9 of Results and point out that multiple different measures may be useful or even necessary to characterize different aspects of consciousness, although we expect the degree of non-separability in a system to be a key measure quantifying its consciousness.”

Reviewer 2 Report

This well-written paper proposes an analysis of consciousness in which the defining characteristic of consciousness is non-separability. Tononi's Integrated Information Theory is an important part of the motivation for this approach: non-separability is meant to introduce an element of "wholeness", in which the whole is more than the sum of the parts. This line of thought is suggestive. However, the paper is very vague on technical details, and seems to vacillate between opposing approaches. I could not distill a coherent story from it. 

Non-separability is a concept coming from quantum mechanics, and the author uses quantum terminology from the beginning (Hilbert spaces, wave functions, and so on). Quantum non-separability goes hand in hand with entanglement and non-locality. But later in the paper the author emphasises that the brain is an effectively classical system, and that we should not think of non-separability as involving entanglement. Moreover, the author points out that the non-separability that he has in mind does not involve non-locality, and he refers to literature in which a classical analogue of quantum non-separability is discussed. This "classical non-separability" can occur in classical field theories, e.g. in superpositions of different polarization states. But this has nothing to do with the wholeness that constituted the initial motivation for the project. Indeed, classical particle systems as discussed by the author never exhibit non-separability in the desired "integrative" form. For example, in the case of the two particles discussed on pp. 9-10, each individual particle is completely characterized by its own properties (position and momentum) and the total system is defined by nothing more than the juxtaposition of the two sets of individual particle properties, There is no non-separability at all, even if we do not use the new coordinates. This is a general feature of classical physics: there is no wholeness encompassing two or more physical systems (as also pointed out in the literature mentioned in the paper). So, it is completely unclear how interactions between classical systems could produce the lack of separability associated with consciousness.

I therefore believe that the basic idea of the paper rests on a technical misunderstanding. 

Author Response

I thank the esteemed reviewer for their comments. I hope they will find that the manuscript is improved now by the addition of new material in response to all reviewers’ points (which can be seen with the Track Changes in the manuscript file). As I understand, the main issue this reviewer saw in the manuscript was the application of the concept of non-separability in the classical case:

“The author points out that the non-separability that he has in mind does not involve non-locality, and he refers to literature in which a classical analogue of quantum non-separability is discussed. This "classical non-separability" can occur in classical field theories, e.g. in superpositions of different polarization states. But this has nothing to do with the wholeness that constituted the initial motivation for the project. Indeed, classical particle systems as discussed by the author never exhibit non-separability in the desired "integrative" form. For example, in the case of the two particles discussed on pp. 9-10, each individual particle is completely characterized by its own properties (position and momentum) and the total system is defined by nothing more than the juxtaposition of the two sets of individual particle properties, There is no non-separability at all, even if we do not use the new coordinates. This is a general feature of classical physics: there is no wholeness encompassing two or more physical systems (as also pointed out in the literature mentioned in the paper). So, it is completely unclear how interactions between classical systems could produce the lack of separability associated with consciousness. I therefore believe that the basic idea of the paper rests on a technical misunderstanding.”

With all respect, I believe there is no technical misunderstanding. The fact that two classical particles in the above-mentioned example are characterized by their respective position and momenta does not mean that such a system is separable. Each degree of freedom can be well-defined by their coordinates in phase space, but their separability has to do with their genuine dependence on each other, such as in how these coordinates evolve in time. If they evolve independently, the system is separable. If evolution of one degree of freedom depends on what happens with others, they are non-separable. That applies to classical and quantum systems. In quantum systems, just like in classical, particles can be identified by their positions or momenta, in the sense that these properties can be measured, and definite answer can be obtained for each particle. In a two-particle quantum entangled system we still can measure the positions of individual particles (or, say, their individual spins). It is just that what happens with one of the particles is linked with what happens with the other, for example for the purposes of time evolution of their possible positions.

Reviewer #3 mentioned an important reference that I now cite in the manuscript, Diaz et al. (Phys. Rev. A, 2022). That paper established classical analogs of quantum measures of non-separability and showed in a few examples how these analogs can be applied to classical systems. One of the examples deals with two coupled harmonic oscillators, and the authors quantified how much non-separability is present in such a system depending on the strength of the coupling. I now describe the main result of their investigation of this example and highlight the dependence of non-separability on the strength of the coupling in Sec. 8. It is a simple example, but it does illustrate exactly the point that a classical two-particle system can very well be non-separable.

I again thank the Reviewer #2 for bringing up this important point, and I added the following text in Sec. 5, which, I hope, clarifies it.

“It also should be noted that, although in the classical case each particle of the system is fully described by its coordinate and momentum (i.e., its coordinate in the phase space), that does not mean that classical systems are automatically separable. Separability has to do with the dependence of degrees of freedom on each other. Like in the example above, if the phase space coordinate of one particle depends on that of another, they are non-separable. This is ultimately reflected in time evolution of the system. Even though we may be able to specify where in phase space each particle is at any moment of time, what matters is whether these phase space coordinates are independent or not. If particles restrict each other’s space of possibilities, they are non-separable to some degree. And as we have seen above, this may depend on the choice of the degrees of freedom (i.e., the choice of basis), either in classical or in quantum case.”

Reviewer 3 Report

In this article, the author introduces the hypothesis that non-separability of degrees of freedom is the fundamental property underlying consciousness in physical systems. The idea, in general, is to me, quite appealing. However, the paper is missing more quantitative results; for example, in Sec. 8, he mentions that the Wigner function offers a convenient framework for describing non-separability- for both quantum and classical systems. But there is not a detailed analysis of this fact inclusive in the case of a Gaussian wave packet where there are precise criteria of separability in the quantum (Phys. Rev.Lett.84(2000)2726, Phys. Rev.Lett.87(2001)2722) and classical (Phys.Rev.A.105(2022)062412) cases. Another point is that the author considers several kinds of separability. Still, he does not mention how to establish a criterion of separability in systems where the separation is other than bipartite. In this case, I consider that the logarithmic negativity (Phys.Rev.A.65(2002)032314, Phys. Rev.A.(2002)042327) could be a very helpful tool, and I  request that the author comment on this subject. Once the author reviews these points, I consider that the article could be published in entropy.

Author Response

I thank the esteemed colleague for the kind words and helpful comments and suggested references. All of that has been useful in improving my manuscript. I included all the references and added material to address specific comments, as follows.

  • “In Sec. 8, he mentions that the Wigner function offers a convenient framework for describing non-separability- for both quantum and classical systems. But there is not a detailed analysis of this fact inclusive in the case of a Gaussian wave packet where there are precise criteria of separability in the quantum (Phys. Rev.Lett.84(2000)2726, Phys. Rev.Lett.87(2001)2722) and classical (Phys.Rev.A.105(2022)062412) cases.”

I added the following text in Sec. 8.

“Notably, Wigner function formalism has been instrumental in establishing separability criteria for physical systems. In particular, it was used to show that in the case of continuous degrees of freedom the so-called Peres-Horodecki criterion provides uncertainty principles that must be obeyed by all quantum separable states, whereas for all bipartite Gaussian states, the Peres-Horodecki criterion is a necessary and sufficient condition of separability [94]. A related criterion for entanglement of a two-part quantum system has been established [95]. Remarkably, Diaz et al. [96] described a classical analog of the quantum covariance matrix and, in the case of Gaussian states, derived expressions for classical analogs of the purity, linear quantum entropy, and von Neumann entropy for classical integrable systems.”

Below that, I added another paragraph describing one of the results from the Diaz et al. paper. It contains formulas and can be better seen in the manuscript.

  • “Another point is that the author considers several kinds of separability. Still, he does not mention how to establish a criterion of separability in systems where the separation is other than bipartite. In this case, I consider that the logarithmic negativity (Phys.Rev.A.65(2002)032314, Phys. Rev.A.(2002)042327) could be a very helpful tool, and I request that the author comment on this subject.”

This is a great point, and I have added the following text in Sec. 9.

“Beyond characterizing non-separability of bi-partite systems, such as can be established using the Von Neumann entropy, it is important to consider compositions other than bi-partite. In this case, logarithmic negativity can be a useful measure [102]. For example, it was shown that the entanglement properties of the ground and thermal states of a closed chain of harmonic oscillators coupled by translationally invariant Hamiltonian can be completely characterized analytically using logarithmic negativity [103].”

Round 2

Reviewer 1 Report

all comments were well addressed

Author Response

Thanks again for the helpful comments at the previous round.

Reviewer 2 Report

I regret that I have to report that the new version of this manuscript has not changed my opinion about its technical soundness. In the beginning of the paper a correct definition of inseparability is given, according to which the total instantaneous state of a many-particle system is not the product of one-particle states. This may occur in quantum mechanics, in the case of entanglement. Contrary to what the authors state, this cannot occur in classical mechanics. Also in contradiction to what the authors suggest in their reply, in the quantum non-separable case the individual particles do not possess their own states. In classical mechanics, however, individual particles always possess their own individual mechanical states in phase space. The example of section 5, and the formula in line 365, are incorrect in this respect. The fact that there is an interaction between particles does not imply that the total instantaneous state is not a product. The new passage starting with line 408 is unfortunately also incorrect: it confuses inseparability with the presence of correlations. This confusion is also visible in other places in the manuscript, where reference is made to the time evolution of the state, even though the definition by the authors of inseparability only refers to instantaneous states. Strangely, there are also passages where the authors themselves warn that inseparability should not be confused with the presence of correlations. Correlations evidently are common in classical mechanics, but they do not introduce holism (holism in the sense that the state of the total system is not equal to the product of the individual particle states).  So, I do not think that the manuscript is technically correct.

Author Response

Perhaps the problem and confusion here have to do with the formulation where I stated that the functions that should be considered are those that describe instantaneous state of the system. I see that this is misleading and agree that one should instead consider functions that describe the time evolution of the system state, and not only its instantaneous state. This is what the functions like the wave function, the Wigner function, and the Liouville density function do.

I therefore changed the main paragraph in Sec. 2 where this is discussed (starting at line 242 in the new version of the manuscript) to read as follows:

“What is the appropriate function f(x_1,x_2,…,t) that should be considered when separability and non-separability of a physical system is in question? Generally speaking, this is the function describing the time evolution of the state of the system, that is, the quantum-mechanical wave function psi(x_1,x_2,…,t) or density matrix, or the equivalents such as the Wigner function [40,41], the Marginal Distribution or Quantum Tomogram [42–46], etc. In a classical world, this can be the function that corresponds to the classical approximation of the wave function (or its equivalents). While any such function describes physical systems equivalently well, we propose below that the Wigner function may offer a particularly fruitful approach, as it has a direct equivalent in the classical case – the Liouville density function in phase space. Note that both in the quantum and classical case this must be a time evolution function of the system (such as, again, wave function, Wigner function, Liouville density function, etc.), rather than just the description of the instantaneous state. This means that it is not only the instantaneous state itself that matters, but the time evolution as well, reflecting the underlying Hamiltonian [32].”

I have also updated Figures 1, 2, and 4, as well as text and formulas throughout the manuscript to reflect this change.

The whole paragraph starting from “It also should be noted that, although in the classical case each particle of the system is fully described by its coordinate and momentum (i.e., its coordinate in the phase space), that does not mean that classical systems are automatically separable” has now been deleted.

The changes done during the previous revision have been accepted, and the new changes for this revision are marked using Track Changes in the manuscript.

Reviewer 3 Report

I am pleased to recommend the article for publication in Entropy. 

Author Response

(The authors gave the same response as above.)

Round 3

Reviewer 2 Report

The core thought of this paper is that consciousness may have a physical basis in non-separability. Non-separability is first defined in the passage starting with line 104; later characterizations, in the paper, repeat the criteria mentioned there. Unfortunately, this definition confuses two completely different notions. First there is the idea that different degrees of freedom are not independent. Dependence between quantities is something that happens across the board, both in quantum and classical physics. For example, the positions variables of the Moon and the Earth are not independent. Does this mean that the Earth-Moon system is more than the sum of its parts, and that we cannot describe it by separately giving the time evolution of the state of the Moon plus the state of the Earth? Of course not. Nevertheless, the authors of the present paper equate lack of independence with wholeness, as characterized in lines 108-114. This is wrong. All the mistakes mentioned in my earlier reports are still present in this latest version. Appeal to the Wigner function does not help: in the classical limit the Wigner function for a single (but composite) system mimics the classical representation by means of a phase space point (see arXiv:quant-ph/0306072 for an explanation). Classical systems never exhibit the wholeness associated with the quantum notion of inseparability; although there may of course be correlations between the quantities of different classical systems. 

I cannot but conclude that the paper is fundamentally flawed. The only interesting way out that I see is to assume that the brain is a quantum system, in which entanglement occurs. In this case the appeal to inseparability could be justified. But the authors make a point of it that the brain is classical, for all practical purposes. 

An alternative escape route would be to claim that the physical substratum of consciousness consists in the presence of correlations between different degrees of freedom. That does not seem a plausible hypothesis though, given that so enormously many degrees of freedom in the inanimate world are correlated.  

Author Response

NA